# Dyslipidemia Exacerbates Meibomian Gland Dysfunction: A Systematic Review and Meta-Analysis

**DOI:** 10.3390/jcm12062131

**Published:** 2023-03-08

**Authors:** Yasufumi Tomioka, Koji Kitazawa, Yohei Yamashita, Kohsaku Numa, Takenori Inomata, Jun-Wei B. Hughes, Rina Soda, Masahiro Nakamura, Tomo Suzuki, Norihiko Yokoi, Chie Sotozono

**Affiliations:** 1Department of Ophthalmology, Kyoto Prefectural University of Medicine, Kyoto 6020841, Japan; 2Buck Institute for Research on Aging, Novato, CA 94945, USA; 3Department of Ophthalmology, Juntendo University Graduate School of Medicine, Tokyo 1130033, Japan; 4Department of Hospital Administration, Juntendo University Graduate School of Medicine, Tokyo 1130033, Japan; 5Department of Digital Medicine, Juntendo University Graduate School of Medicine, Tokyo 1130033, Japan; 6AI Incubation Farm, Juntendo University Graduate School of Medicine, Tokyo 1130033, Japan; 7Graduate School of Engineering, University of Tokyo, Tokyo 1138656, Japan; 8Department of Ophthalmology, Kyoto City Hospital Organization, Kyoto 6048845, Japan

**Keywords:** hyperlipidemia, aging, senescence, inflammation, hypercholesterolemia, hypertriglyceridemia, MGD

## Abstract

Dry eye is a multifactorial and common age-related ocular surface disease. Dyslipidemia has been reported to be involved in meibomian gland dysfunction (MGD). However, it has not been clearly identified which lipid abnormality is responsible for MGD. In this systematic review and meta-analysis, we discuss how lipid profile changes with aging is responsible for MGD development. Methods. An article search was performed in PubMed, EMBASE, and Web of Science. Eleven studies involving dyslipidemia in patients with MGD were identified. Five out of eleven studies were further analyzed with meta-analysis. The preferred reporting items for Systematic Reviews and Meta-Analyses (PRISMA) reporting guidelines were followed. Study-specific estimates (prevalence of dyslipidemia in MGD patients) were combined using one-group meta-analysis in a random-effects model. Results. Meta-analysis revealed that high total cholesterol (TC) and high triglycerides (TG) were significantly associated with MGD prevalence, with odds ratios of 5.245 (95% confidence interval [CI]: 1.582–17.389; *p* < 0.001) and 3.264 (95% CI: 1.047–10.181; *p* < 0.001), respectively, but high low-density lipoprotein (LDL) and low high-density lipoprotein (HDL) were not identified. Systematic review found that the percentage of MGD patients with TC ≥ 200 mg/dL ranged from 20.0–77.6%, TG ≥ 150 mg/dL ranged from 8.3–89.7%, whereas, in the aged-match-adjusted controls, TC range of 200 mg/dL or higher and TG range of 150 mg/dL was 6.1–45.1% and 1.1–47.8%, respectively. The severity of MGD was higher with dyslipidemia. Conclusion. Dyslipidemia and higher TC and TG are significant risk factors for MGD.

## 1. Introduction

Dry eye is a multifactorial age-related ocular surface disease with a high prevalence of disease in the world, ranging from 5 to 50% [1,2,3]. Dry eye is characterized by the instability of tear film on the ocular surface, leading to the various symptoms that influence the productivity at the workplace due to the lower quality of vision [4]. The ocular surface is a complex anterior part of the eye comprised by corneal/conjunctival cells, lacrimal glands, and meibomian glands (MGs). MGs produce lipids and secrete meibum which is essential to create the outer most lipid layer of the tear film (TF) to stabilize it. Thus, meibomian gland dysfunction (MGD) has been considered to be the leading cause of evaporative dry eye as a result of deficiency of the lipid layer.

The prevalence of MGD varies widely from 3.6% to 68% among those aged 40 years and older [5,6,7,8,9,10,11,12,13]. Dyslipidemia is manifested by elevation of the total cholesterol (TC), increased triglycerides (TGs) and low-density lipoprotein (LDL) levels, and decreased high-density lipoprotein (HDL) levels. Plasma levels of TC and LDL are well known to increase with normal aging [14,15], while HDL declines with age [16]. Dyslipidemia can also be caused by diet and hormonal changes associated with menopause [16,17,18]. Dyslipidemia has recently been reported to be one of the risk factors to develop MGD [19]. However, the type of plasma lipids associated with MGD has varied among studies and it has not been clearly identified which plasma lipid abnormality is responsible for MGD.

In this study, we conducted a systematic review and meta-analysis on the association between dyslipidemia and MGD to determine which plasma lipid abnormalities are associated with the development of MGD. We also investigated the relationship between the severity of MGD and dyslipidemia, and discussed how plasma lipid profile with aging is involved in MGD.

## 2. Materials and Methods

### 2.1. Database Retrieval and Search Strategy

We searched published research papers (from inception to May 2022) using electronic bibliographic databases including PubMed, EMBASE, and Web of Science. We followed the guidelines of the preferred reporting items for systematic reviews and meta-analyses (PRISMA) protocols [20]. Publications related to MGD and dyslipidemia were searched with two search filters. The first search filter included the following terms combined with an “OR“: “hyperlipidemia”, “dyslipidemia” “total cholesterol” “low-density lipoproteins” “high-density lipoproteins” “triglycerides”. The second search filter included the following terms combined with “OR”: “meibomian gland dysfunction”, and “MGD”. Two filters were combined with an “AND”. The search literatures were narrowed down according to the search strategy summarized in a flowchart (Figure 1).

### 2.2. Inclusion and Exclusion Criteria

The study inclusion and exclusion criteria are presented in Table 1. The systematic review included papers in English and full articles accessible with suitable topics. Integral data were collected for meta-analysis. Mouse studies, experimental methods, systematic reviews, reviews and preprinted articles were excluded.

### 2.3. Data Extraction

Search results were compiled using EndNote X9.3.3 (Clarivate Analytics, Philadelphia, PA, USA). To adhere to the quality standards for reporting systematic reviews and meta-analyses of observational studies [21], two reviewers (Y.T. and K.K.) independently agreed on the selection of studies that meet the criteria and reached consensus on the studies to be included. The same two reviewers independently assessed the full text of the records that were deemed eligible in consensus. Disagreements between the reviewers regarding the extracted data were resolved through discussion with a third reviewer (Y.Y.). The following data were extracted: first author name, year of publication, prevalence of dyslipidemia, plasma level of TC, TG, LDL, and HDL. The unit of analysis was prevalence of dyslipidemia in patients with or without MGD.

### 2.4. Risk-of-Bias Assessment

Two investigators (Y.T. and K.K.) independently assessed the risk of bias in each included study. We used the tool proposed by Hoy et al. [22], and resolved discrepancies during the risk-of-bias assessment by discussion or adjudication by a third investigator (Y.Y.) where needed.

### 2.5. Assessment of Study Quality

Two investigators (Y.T. and Y.Y.) independently assessed the study quality in each included study. The Newcastle–Ottawa Quality scale (NOS) [23] for assessing the quality of nonrandomized studies in meta-analyses was used to evaluate the quality of included studies; a score of 7 or higher out of 9 on the NOS scale indicates high quality.

### 2.6. Statistical Analysis

A meta-analysis was performed using OpenMetaAnalyst version 12.11.14 (available from http://www.cebm.brown.edu/openmeta/, accessed on 27 May 2022) [24,25]. As there was no heterogeneity (I2 < 50%) among the studies, a fixed-effects model was applied for meta-analysis. When I2 was >50%, a random-effect model was used.

## 3. Results

### 3.1. Study Selection

The database search yielded 40 articles. After removing 17 duplicates, 23 articles were reviewed based on the title and abstract. After removing 12 papers which did not meet the eligibility requirement (i.e., no raw data to perform statistical analysis), 11 papers were selected for full-text evaluation and deemed eligible for the systematic review. Among these articles, five articles were investigated in the meta-analysis [26,27,28,29,30] (Figure 1).

### 3.2. Study Characteristics

Eleven studies [26,27,28,29,30,31,32,33,34,35,36] in this systematic review and five studies [26,27,28,29,30] in the meta-analysis were published between 2010 and 2021. Among the included studies, three were from India, two were from United states, and the remaining eight were from Italy, Iran, Taiwan, Nepal, Saudi Arabia, and Korea. The sample size ranged from 30 to 4700 participants. Age of participants was 15 to 86 years old. As for diagnostic criteria, eight studies [26,27,28,29,30,33,34,36] were based on the report submitted by the International Workshop on Meibomian Gland Dysfunction and Management in 2011 [37,38,39], three studies [31,32,35] were based on the Foulks and Bron report [40]. All the five studies for meta-analysis followed the diagnosis criteria based on the international workshop of MGD [37,38,39] and used age-matched patients without MGD in each study as a control. Risk of bias assessment revealed that all the papers included in this meta-analysis were judged to be of low risk (Table 2).

### 3.3. Quality of Evidence Assessment

All the five studies for meta-analysis were assessed by NOS: four case–control and cohort studies were of high quality and one was of moderate quality, with a mean NOS score of 7.2 (SD 1.09) (Table 3).

### 3.4. Risk of MGD in Dyslipidemia Patients

We examined the prevalence of dyslipidemia in MGD patients according to each lipid component [26,27,28,29,30,31,32,33,34,35,36]. The results showed that the percentage of MGD patients with TC ≥ 200 mg/dL ranged from 20.0–77.6%, TG ≥ 150 mg/dL ranged from 8.3–89.7%, LDL ≥ 130 mg/dL ranged from 17.2–39.1%, and HDL was 40 mg/dL or less in 3.8–56.7% of patients (Table 4). In the age-match-adjusted controls, TC was 200 mg/dL or higher in 6.1–45.1%, TG was 150 mg/dL or higher in 1.1–47.8%, LDL was 130 mg/dL or higher in 3.4–32.8%, and HDL was 40 mg/dL or lower in 1.7–53.3% of the age-match-adjusted controls (Table 5).

We next examined the dyslipidemia values in the MGD patients [27,28,29,32,33]. The mean TC range was 186–213 mg/dL, TG range was 90–188 mg/dL, LDL range was 105–135 mg/dL, and HDL range was 43–62 mg/dL (Table 5). Controls had a mean TC range of 157–198 mg/dL, TG range of 74–135 mg/dL, LDL range of 92–119 mg/dL, and HDL range of 46–53 mg/dL (Table 5).

### 3.5. Meta-Analysis of Risk of Dyslipidemia in MGD Patients

A total of five studies [26,27,28,29,30] assessed dyslipidemia in MGD patients compared to controls. Five studies involving 421 participants with MGD and 402 participants without MGD were included in the meta-analysis and the diagnosis of MGD used criteria based on the international workshop of MGD [37,38]. The confounder-adjusted results from five studies revealed that there was significant association between MGD and high TC [overall OR = 5.245; 95% CI: 1.582–17.389; *p* < 0.001; I^2^ = 8898%] and high TG [overall OR = 3.264; 95% CI: 1.047–10.181; *p* < 0.001; I^2^ = 8013%] (Figure 2). However, there was no significant association between MGD and high LDL [overall OR = 3.429; 95% CI: 1.836–6.403; *p* = 0.106; I^2^ = 5092%] and low HDL [overall OR = 1.018; 95% CI: 0.649–1.597; *p* = 0.459; I^2^ = 0%] (Figure 2).

### 3.6. MGD Severity and Dyslipidemia

Five out of eleven studies were further analyzed to investigate how dyslipidemia associates with the severity of MGD [26,28,30,34,35]. Irfan KSA [34], Banait S [30], and Guliani BP [28] assessed the severity of MGD based on meibum quality on a 0–3 scale (clear, cloudy, cloudy meibum with debris, and thick toothpaste-like meibum), expressibility of meibum on a scale of 1 to 3 (3–4 glands expressible, 1–2 glands expressible, and no glands expressible) and corneal and conjunctival staining scores on a range of 0–15. Bukhari AA [26] defined the severity of MGD as the presence of ductal plugging and expression of clear meibum at Grade 1, the presence of ductal plugging and expression of cloudy meibum at Grade 2, and the presence of ductal plugging and inspissated material and the lack of meibum expression or glandular loss at Grade 3. Tulsyan N [35] assessed MGD based on the symptoms and signs of MGD: blink rate less than 15/min, tear film break-up time less than 10 sec, Schirmer test reading from normal to severe (normal ≥ 15 mm, mild: 0–15 mm, moderate: 5–10 mm, severe:< 5 mm). Tulsyan also evaluated the severity of MGD based on the above-mentioned criteria as Grade 0 (Normal), Grade 1 (Subclinical), Grade 2 (Minimal), Grade 3 (Mild), Grade 4 (Moderate) and Grade 5 (Severe). Overall, the number of patients with dyslipidemia, including high TC, high TG, high LDL, and low HDL, increased with the severity of MGD unlike those who had no MGD (Figure 3).

## 4. Discussion

Altered lipid composition of MGs is observed in patients with MGD [41,42]. There have been several studies on the association between dyslipidemia and MGD [26,27,28,29,30,31,32,33,34,35,36,43,44,45], but sample size in each study was limited. This systematic review and meta-analysis revealed that high TC and TGs were significantly associated with MGD, with odds ratios of 5.245 and 3.264, respectively, but high LDL and low HDL were not identified to be associated with MGD. Additionally, it became clear that dyslipidemia contributed not only to the prevalence but also to the severity of MGD, in particular, TC and TG levels. Dao et al. did report that there was a greater number of MGD patients with high TC and smaller number of MGD patients with high TGs [31]; however, they used historical data as a control, indicating that their diagnosis criteria may differ from our meta-analysis.

Many mouse models have also shown that hyperlipidemia is associated with MGD [46,47]. Apolipoprotein E (ApoE) knockout mice, which are characterized by hypercholesterolemia and hypertriglyceridemia [48], displayed an MGD phenotype, including MG dropout, abnormal MG acinar morphology, dilated MG duct and plugging of the MG orifice [46]. Another mouse model with dyslipidemia is the high-fat diet (HFD) mice which show increased plasma lipid levels, resulting in larger MG area and an increase in saturated lipid species [47]. However, it is debated whether dyslipidemia directly affects lipid biosynthesis in meibocytes of MG. Lipid biosynthesis in the MGs is organized in a complex network, which was termed meibogenesis by Butovich IA [49]. Although a lot of enzymes that biosynthesize lipids in MGs have been studied, it is also possible that plasma lipids are involved in meibogenesis. Dietary cholesterol and TGs with labeled tracers allowed exploration into the dynamics of their accumulation in meibum and were subsequently found to be directly incorporated into meibum [50], suggesting that plasma lipids can influence the meibogenesis in MGs. It should be noted that Ha M et al. reported that dyslipidemia did not influence MGD subtypes [33], and Butovich IA and Suzuki T reported that abnormal meibum, tears and sebum (AMTS) patients displayed no abnormalities of plasma lipid [51]. Composition of lipids in the meibum of the patients with MGD is variable [41,42,51,52], so further study is needed to conclusively determine the role of plasma lipids in the composition of meibum.

Although the change of MGs due to dyslipidemia is well characterized, the change of MGs may also be associated with age-related alterations regardless of the plasma lipid profile. We recently performed a systematic review and meta-analysis to see how aging affects the ocular surface, and reported that the aging process significantly alters the microenvironment at the ocular surface, including MGs [53]. The aging process pathologically alters MGs, resulting in the decrease of the density of MGs and an increase of the gland drop-out and obstruction rates of MGs [54,55,56,57,58], possibly due to decreased levels of peroxisome proliferator-activated receptors (PPAR)-γ [59,60]. Furthermore, lipid composition in meibum is reportedly altered with age. Non-polar lipids significantly decreased whereas polar lipids significantly increased in both elderly and MGD patients [41]. However, in the same publication, TGs significantly increased only in MGD patients [41], suggesting that TGs are the only lipid group quantitatively altered in MGD. In addition, intracrine activity in MGs decreased with age, leading to MGD by decreased meibomian oil production [58]. Thus, the ocular surface seems to deteriorate more in elderly individuals, especially ones with MGD. However, it is not yet clear how aging and dyslipidemia change lipid composition to exacerbate detrimental changes to the ocular surface.

Our present meta-analysis revealed that the lipid profile was altered in MGD patients. In addition, the lipid profile is also altered with age as plasma TG level has been shown to be elevated with age [14]. To date, many studies on dyslipidemia have reported that the level of TC and TGs increases progressively with normal aging [15,17,61,62], and TGs have been proposed as a biomarker of aging. Our present systematic review showed that high TC and TGs was a significant risk to the development and severity of MGD, suggesting that MGD is more common at older ages and severe stages of MGD are associated with aging. TG levels are influenced by daily diet as the most abundant dietary lipids are TGs [18]. Thus, having good dietary habits can be a reasonable prevention strategy for MGD. In fact, Siak JJK et al. reported that medication with fibrates, which are agonists of the nuclear receptor PPAR-α and reduce plasma TGs, reduced the risk of MGD development [43]. TC also increases with age but declines at older ages [63,64], suggesting that TCs may not continually be involved in MGD development, unlike TGs. The level of LDL also increases with age, but increases are more prevalent in men than women until middle ages [65]. Between ages 50–60, LDL is elevated in women, but stays relatively constant for men. As both men and women reach ages above 60, though, the LDL levels plateau [62].

Two reports from Guliani [28] and Tulsyan [35] summarized the MGD severity with aging. Gulian et al. reported that there was a strong association between age and severity of MGD [0.582, 95%CI (0.426–0.704), *p*-value < 0.0001] [28], and Tulsyan et al. demonstrated that the severity of MGD increased with aging [35], indicating that the increased age of patients led to more severe MGD. The link between aging and changes in MG, as well as aging and the severity of MGD, could be explained by the accumulation of senescent cells in the MG, as recently, it is known that senescent cells accumulate with age and it has recently been reported that hyperlipidemia induces cellular senescence [66]. Senescent cells accumulating in the MG of both mice and humans can be seen by the decrease in Ki67-positive cells with age [53,56,57,59]. Senescent cells acquire a complex, often pro-inflammatory, secretory phenotype termed the senescence-associated secretory phenotype (SASP) [67], which can change the neighboring microenvironment, such as by lipid-induced inflammation. There is evidence that excessive calorie intake allows adipocytes to undergo senescence and cause an inflammatory response which leads to atherosclerosis and heart failure [68]. Thus, an abnormal profile of lipid may accelerate aging though the dyslipidemia-induced cellular senescence in MGs, which results in MGD. The present meta-analysis found that LDL was not identified to be elevated in the patients with MGD. However, it may be possible that an increase of LDL levels up to middle age induces cellular senescence, which then permanently changes the microenvironment in MG without any further increases in LDL, and finally leads to MGD at older ages.

A few limitations have to be addressed. First, it is unclear whether MGD is caused by dyslipidemia or whether it is associated with simply hyperlipidemia, and there is no evidence that dyslipidemia is the essential cause of MGD. Previous mouse studies have reported an increased incidence of MGD in mice with altered lipid abnormalities [47,48], but whether this translates to humans needs to be carefully examined. Second, the definition of MGD among reports should also be carefully examined because the classification and definition of MGD was not determined until the international workshop on MGD in 2011 [37,38,39], which unified a common diagnosis worldwide. All of the papers included in the present meta-analysis were evaluated using the same criteria according to the International Workshop on Meibomian Gland Dysfunction and Management in 2011 [37,38,39], suggesting that the bias among reports on MGD should be minimized. Third, as there was a lack of randomized control studies and epidemiological studies on the detailed lipid profile associated with MGD, meta-analysis in this study was performed based on reports from case-series. We need high evidence information based on a large population-study to make a conclusion. Fourth, the meta-analysis did not consider these MGD subtypes: hyposecretory, obstructive, and hypersecretory MGD. Ha M et al. reported that there was no statistically significant difference in lipid profile levels between any of the MGD subtypes listed above [33]. Lastly, our systematic review was not registered in PROSPERO. Further study is needed.

## 5. Conclusions

In conclusions, the present systematic review and meta-analysis showed that among dyslipidemias, high TC and TG were significant risk factors for the development of MGD, and abnormal lipid profile was associated with the risk of severe stage of MGD, suggesting that age-related dyslipidemia may be involved in the development of MGD. These findings cannot yet determine if dyslipidemia is cause or effect; however, further clarity on whether lipids are involved in the pathogenesis or progression of MGD may be able to prevent or treat MGD through therapeutic interventions based on the subtype of dyslipidemia.

## Figures and Tables

**Figure 1 jcm-12-02131-f001:**
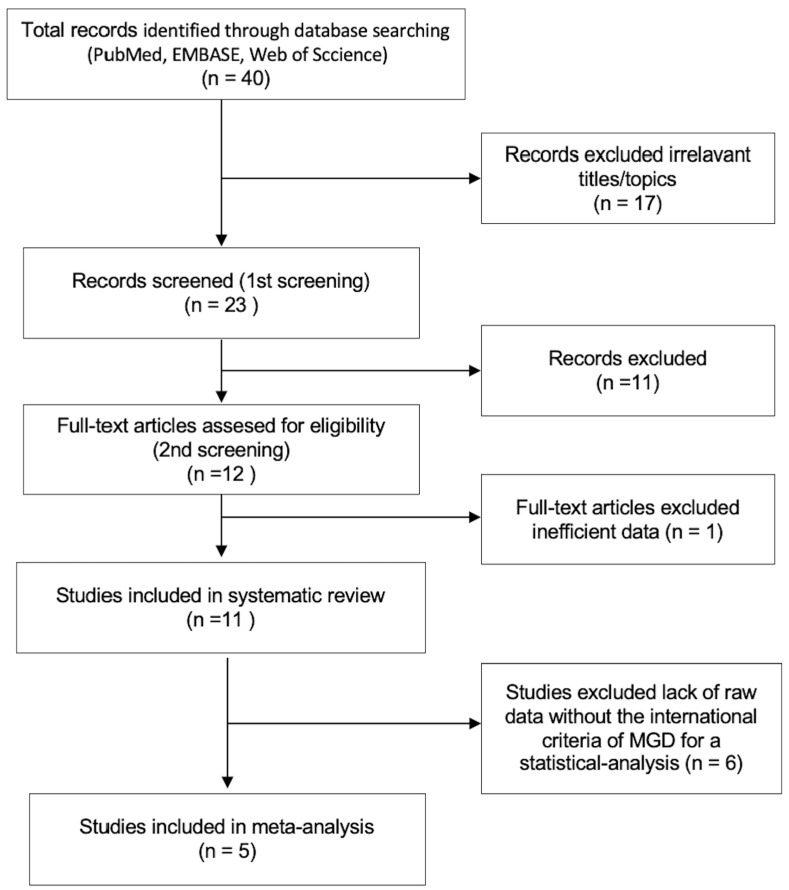
Flow chart of study selection for systematic review and meta-analysis.

**Figure 2 jcm-12-02131-f002:**
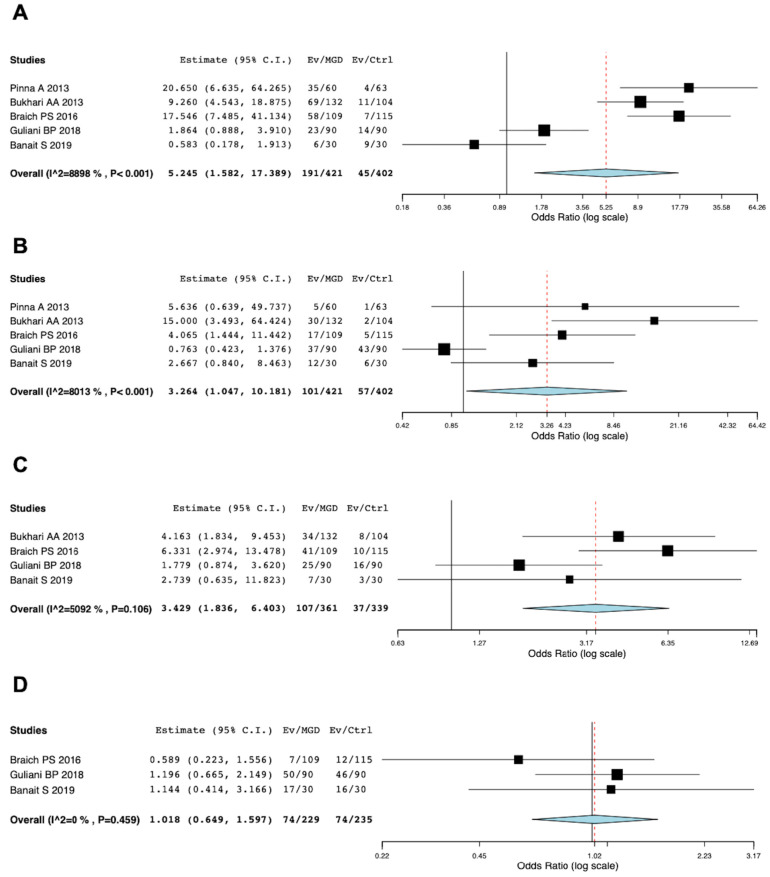
Meta-analysis for the presence of dyslipidemia in MGD patients. (**A**). Total cholesterol (TC) ≥ 200 mg/dL, (**B**). Triglyceride (TG) ≥ 150 mg/dL, (**C**). Low-density lipoprotein (LDL) ≥ 150 mg/dL, (**D**). High-density lipoprotein (HDL) ≤ 40 mg/dL. References [26,27,28,29,30] are cited in the figure.

**Figure 3 jcm-12-02131-f003:**
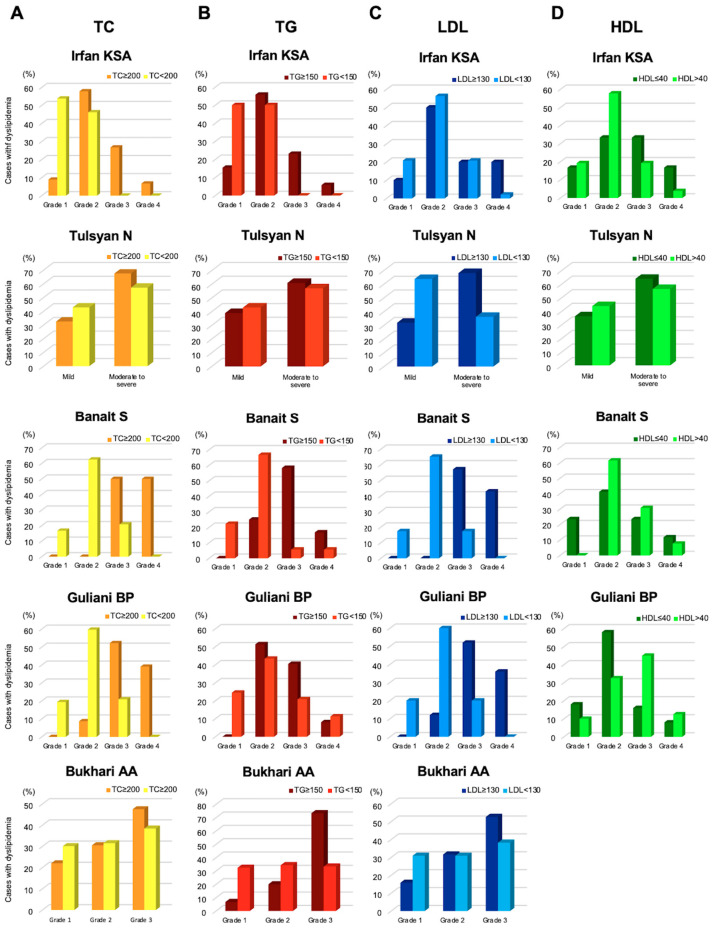
Severity of MGD and dyslipidemia. (**A**). Cases with total cholesterol (TC) ≥ 200 mg/dL according to the severity of MGD, (**B**). Triglyceride (TG) ≥ 150 mg/dL, (**C**). Low-density lipoprotein (LDL) ≥ 150 mg/dL, (**D**). High-density lipoprotein (HDL) ≤ 40 mg/dL. Reproduced from [34] and [35], with permission from Ubiquity Press, 2023, [28] with permission from Medknow Publications, 2023, and [26] with permission from Wolters Kluwer Health, Inc, 2023.

**Table 1 jcm-12-02131-t001:** Inclusion and exclusion criteria.

Inclusion Criteria
1. Study objective: Potential involvement of dyslipidemia in the MGD were investigated.
2. Study design: Retrospective (cross-sectional studies, case-control studies, case series, and case reports) and prospective studies
3. Outcome: The level of TC, TG, LDL, and HDL in the patients with or without MGD or based on the severity of MGD
Exclusion Criteria
1. Clinical guidelines, consensus documents, reviews, and conference proceedings
2. Animal-based study
3. Studies not involved in the study objective
4. Lack of data to analyze
5. Pre-printed articles
6. Articles not published in English

MGD, meibomian gland dysfunction; TC, total cholesterol; TG, triglyceride; LDL, low-density lipoprotein; HDL, high-density lipoprotein.

**Table 2 jcm-12-02131-t002:** Risk of bias assessment for the papers in the meta-analysis.

Risk of Bias Domains	Bukhari, 2013 [26]	Pinna, 2013 [29]	Braich, 2016 [27]	Guliani, 2018 [28]	Banait, 2019 [30]
Representativeness of target population to national population	N	N	N	N	N
Representativeness of sampling frame to target population	Y	Y	Y	Y	Y
Sampling: random or census	N	N	N	N	N
Minimal non-response bias	Y	Y	Y	Y	Y
Data collected directly from participants	Y	Y	Y	Y	Y
Acceptable case definition	Y	Y	Y	Y	Y
Valid and reliable instrument	Y	Y	Y	Y	Y
Same mode of data collection for all participants	Y	Y	Y	Y	Y
Prevalence period appropriate	Y	Y	Y	Y	Y
Numerator(s) and denominator(s) appropriate	Y	Y	Y	Y	Y
Overall risk of bias	LHI	L	L	L	L

Y—yes; N—no; L—low.

**Table 3 jcm-12-02131-t003:** Quality of evidence assessment for the papers in the meta-analysis.

Risk of Bias Domains	Bukhari, 2013 [26]	Pinna, 2013 [29]	Braich, 2016 [27]	Guliani, 2018 [28]	Banait, 2019 [30]
Selection Overall risk of bias	****	***	***	**	***
Comparability H—high quality;	**	**	**	*	**
Exposure	***	**	**	***	**
Overall risk of bias	H	H	H	I	H

H—high quality; I—intermediate. The number of * indicates the score of NOS scale.

**Table 4 jcm-12-02131-t004:** Summary of dyslipidemia in patients with or without meibomian gland dysfunction.

First Author	Year	Country	MGD	Participants (n)	Age (Mean)	TC ≥200 mg/dL	TG ≥ 150 mg/dL	LDL ≥ 130 mg/dL	HDL ≤ 40 mg/dL
Dao AH [31]	2010	United States	(+)	46	27–82 * (52)	67.4%	15.2%	39.1%	6.5%
			(−)	historical control	20–65 over (47)	45.1%	33.1%	32.8%	15.7%
Pinna A [29]	2013	Italy	(+)	60	18–54 (38)	58.3%	8.3%	-	-
			(−)	63	18–54 (36)	6.3%	1.1%	-	-
Bukhari AA [26]	2013	Saudi Arabia	(+)	132	15–78 **(49)	52.3%	22.7%†	25.8%	3.8%
			(−)	104		10.6%	1.9%	7.3%	-
Braich PS [27]	2016	United States	(+)	109	20–72 (47)	53.2%	15.6%	37.6%	6.4%
			(−)	115	19–75 (46)	6.1%	4.3%	8.7%	10.4%
Chen A [32]	2017	Taiwan	(+)	89	30–60 (49)	The number of patients with dyslipidemia was not described			
			(−)	199	30–60 (49)	The number of patients with dyslipidemia was not described			
Hashemi H [36]	2017	Iran	(+)	4700	45–69 (55)	OR:1.000, 95%CI (1.000–1.002)	OR:1.0001, 95%CI (1.000–1.008)	-	OR:0.99, 95%CI (0.986–0.998)
Guliani BP [28]	2018	India	(+)	90	18–54	25.6%	41.1%	27.8%	55.6%
			(−)	90	18–54	15.6%	47.8%	17.8%	51.1%
Banait S [30]	2019	India	(+)	30	31–80 (62)	20.0%	40.0%	23.3%	56.7%
			(−)	30	31–80	30.0%	20.0%	10.0%	53.3%
Ha M [33]	2021	South Korea	(+)	95	19–86 (58)	41.1%	25.2%	20.0%	5.3%
			(−)	475	19–80 (56)	The number of patients with dyslipidemia was not described			
Irfan KSA [34]	2021	India	(+)	58	18–65 (49)	77.6%	89.7%	17.2%	10.3%
			(−)	58	18–65 (49)	19.0%	43.1%	3.4%	1.7%
Tulsyan N [35]	2021	Nepal	(+)	400	31-	20.0%	57.0%	23.5%	40.0%

MGD, Meibomian gland dysfunction; TC, total cholesterol; TG, triglyceride; LDL, low-density lipoprotein; HDL, high-density lipoprotein; OR, odds ratio; CI, confidence interval; * Age of all participants; ** Age of all participants including controls; †TG ≥ 200 mg/dL.

**Table 5 jcm-12-02131-t005:** Lipid profile among patients with or without meibomian gland dysfunction.

First Author	Year	Country	MGD	Participants (n)	TC (mg/dL)	TG (mg/dL)	LDL (mg/dL)	HDL (mg/dL)
Pinna A [29]	2013	Italy	(+)	60	210 (4)	90 (47)	128 (4)	62 (2)
			(−)	63	163 (3)	74 (27)	94 (3)	53 (1)
Braich PS [27]	2016	United states	(+)	109	203 (13)	99 (42)	126 (10)	53 (4)
			(−)	115	157 (15)	82 (37)	92 (12)	46 (3)
Chen A [32]	2017	Taiwan	(+)	89	213 (34)	188 (109)	135 (30)	48 (12)
			(−)	199	188 (35)	130 (74)	108 (30)	51 (13)
Guliani BP [28]	2018	India	(+)	90	186 (59)	150 (63)	116 (50)	43 (21)
			(−)	90	N/A	N/A	N/A	N/A
Ha M [33]	2021	South Korea	(+)	95	193 (34)	128 (75)	105 (31)	62 (13)
			(−)	475	198 (38)	135 (90)	119 (34)	51 (12)

MGD, Meibomian gland dysfunction; TC, total cholesterol; TG, triglyceride; LDL, low-density lipoprotein; HDL, high-density lipoprotein; Values are expressed as average (standard deviation).

## Data Availability

All data generated or analyzed during this study are included in this published article.

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
