# Peer review of "Dyslipidemia Exacerbates Meibomian Gland Dysfunction: A Systematic Review and Meta-Analysis"

_jcm, 2023, doi:10.3390/jcm12062131_

Round 1
Reviewer 1 Report
1.Only 5 papers were investgated in the meta-analysis,the number of included studies was relatively small. If possible,search more databases.
2.The papers published from inception to May 2022. During such long time, did MGD have unified diagnostic standards,diagnostic methods and inspection methods? Those would affect analysis results.
Author Response
Comment 1: Only 5 papers were invesitgated in the meta-analysis, the number of included studies was relatively small. If possible,search more databases.
Response: Thank you for your cogent review and suggestions. We searched PubMed, EMBASE, Web of Science, but determined that there were the only five articles with clear numerical values that meet our inclusion and exclusion criteria for meta-analysis.
Comment 2: The papers published from inception to May 2022. During such long time, did MGD have unified diagnostic standards,diagnostic methods and inspection methods? Those would affect analysis results.
Response: International diagnostic criteria for MGD were defined at the 2011 International Workshop on MGD, and prior to that, there were no unified diagnostic criteria, diagnostic methods, or testing methods. As the reviewer pointed out, unified diagnostic standards would affect the analysis results. However, since the papers included in this systematic review are from 2010 or later and the meta-analysis from 2013 or later, the diagnostic criteria, diagnostic methods, and testing methods for MGD have been standardized. So, we consider that there will be little impact on the results of the analysis. Please note that we have described a little bit about this limitation in the Limitation section (Page 10)
Reviewer 2 Report
1) Relationship between dyslipedemia and MGD has been studied extensively
2)This study doesnt really add anything new to our existing knowledge.
3)Could have added results of studies on the relationship between the subtypes MGD and Dyslipidemia.
Author Response
Comment 1: Relationship between dyslipedemia and MGD has been studied extensively
Response: Thank you for your cogent review and suggestions. Although there are several studies on the association between dyslipidemia and MGD, this meta-analysis newly confirmed that among dyslipidemias, high TC and high TG were significantly associated with MGD. In this paper, high TC and high TG were found to be deeply associated with MGD, which we consider is a new finding.
Comment 2: This study doesn’t really add anything new to our existing knowledge.
Response: As we described the Response above, this meta-analysis revealed that high TC and high TG were significantly associated with MGD, which we consider was a new finding. In addition, the number of patients with dyslipidemia, including high TC, high TG, high LDL, and low HDL, increased with the severity of MGD unlike those who had no MGD. These findings suggest that more impairment of lipid profile is closely related to the onset and severity of MGD.
Comment 3: Could have added results of studies on the relationship between the subtypes MGD and Dyslipidemia.
Response: Thank you for your cogent review and suggestions. Subtype classification was not able to be evaluated because it was not mentioned in any of the papers included in the meta-analysis, although it was noted that there was no difference in subtype classification in the paper by Ha et al. We have described a little bit about the relationship between the subtype of MGD and dyslipidemia in the Limitation section (Page 10-11).
Reviewer 3 Report
In this systematic review and meta-analysis, Tomioka et al. investigated how lipid profile changes with aging is responsible for MGD development. The manuscript is very interesting and well-written; however, there are some issues which need do be addressed, as outlined below.
- I recommend the Authors use the Newcastle-Ottawa Scale (NOS) to assess the quality of each study (Wells G et al. The Newcastle-Ottawa Scale (NOS) for assessing the quality of non-randomised studies in meta-analyses. 2013. Available at: http://www.ohri.ca/programs/clinical_epidemiology/oxford.asp). The NOS evaluates the following components: selection of the cohort, comparability of cohorts on the basis of the design or analysis, how the exposure was ascertained, and how the outcomes of interest were assessed. NOS scores of 1-3, 4-6, 7-9 indicate low, intermediate, and high quality, respectively.
Author Response
Comment 1: I recommend the Authors use the Newcastle-Ottawa Scale (NOS) to assess the quality of each study (Wells G et al. The Newcastle-Ottawa Scale (NOS) for assessing the quality of non-randomised studies in meta-analyses. 2013. Available at: http://www.ohri.ca/programs/clinical_epidemiology/oxford.asp). The NOS evaluates the following components: selection of the cohort, comparability of cohorts on the basis of the design or analysis, how the exposure was ascertained, and how the outcomes of interest were assessed. NOS scores of 1-3, 4-6, 7-9 indicate low, intermediate, and high quality, respectively.
Response: Thank you for your cogent review and suggestions. We have assessed the quality of this study according to the Newcastle-Ottawa Scale (NOS) and added the NOS in the Methods and Results section(page 3, page 5 and Table 3).